# Near-Infrared Reflectance Spectrophotometry (NIRS) Application in the Amino Acid Profiling of Quality Protein Maize (QPM)

**DOI:** 10.3390/foods11182779

**Published:** 2022-09-09

**Authors:** Emmanuel Oladeji Alamu, Abebe Menkir, Michael Adesokan, Segun Fawole, Busie Maziya-Dixon

**Affiliations:** 1Food and Nutrition Sciences Laboratory, International Institute of Tropical Agriculture, Southern Africa Research and Administration Hub (SARAH) Campus, Lusaka 10101, Zambia; 2Food and Nutrition Sciences Laboratory, International Institute of Tropical Agriculture (IITA), Ibadan 20001, Nigeria; 3Maize Breeding Unit, International Institute of Tropical Agriculture (IITA), Ibadan 20001, Nigeria

**Keywords:** NIRS, amino acids, quality protein maize, HPLC, screening, model, calibration, validation

## Abstract

The accurate quantification of amino acids in maize breeding programs is challenging due to the high cost of analysis using High-Performance Liquid Chromatography (HPLC) and other conventional methods. Using the Near-Infrared Spectroscopic (NIRS) method in breeding to screen many genotypes has proven to be a fast, cost-effective, and non-destructive method. Thus, this study aimed to develop and apply the NIRS prediction models for quantifying amino acids in biofortified quality protein maize (QPM). Sixty-three (63) QPM maize genotypes were used as the calibration set, and another twenty (20) genotypes were used as the validation set. The microwave hydrolysis system coupled with post-column derivatization with 6-amino-quinoline-succinimidyl-carbamate as the derivatization reagent and the HPLC method were used to generate the reference data set used for the calibration development. The calibration models were developed for essential and non-essential amino acids using WINSI Foss software. Good coefficients of determination in calibration (R^2^_cal_) of 0.91, 0.93, 0.93, and 0.91 and low standard errors in calibrations (SEC) of 0.62, 0.71, 0.26, and 1.75 were obtained for glutamic acids, alanine, proline, and leucine, respectively, while aspartic acids, serine, glycine, arginine, tyrosine, valines, and phenylalanine had fairly good R^2^_Cal_ values of 0.86, 0.71, 0.81, 0.78, 0.68, 0.79, and 0.75. In contrast, poor (R^2^_cal_) was obtained for histidine (0.07), cystine (0.09), methionine (0.09), lysine (0.20), threonine (0.51), and isoleucine (0.09), respectively. The models’ prediction performances (R^2^_pred_) and standard error of prediction (SEP) were reasonably good for certain amino acids such as aspartic acid (0.90), glycine (0.80), arginine (0.94), alanine (0.90), proline (0.80), tyrosine (0.83), valine (0.82), leucine (0.90), and phenylalanine (0.88) with SEP values of 0.24, 0.39,0.24, 0.93, 0.47,0.34, 0.78, 2.20, and 0.77, respectively. However, certain amino acids had their R^2^_pred_ below 0.50, which could be improved to become useful for screening purposes for those amino acids. Further work is recommended by including a training set representing the sample population’s variance to improve the model’s performance.

## 1. Introduction

Maize (*Zea mays* L.) is a cereal crop that has found application in both industrial and non-industrial sectors. It is the third major cereal crop in the world after wheat and rice. It is used for livestock feed and human consumption and supplies about one-fifth of the total daily calories [1]. Amino acids are building blocks for protein in the body of monogastric animals, including man. Some are essential because organisms do not synthesize them; hence, they must be supplied as part of the organism’s diet. Examples include lysine, tryptophan, and valine. However, non-essential amino acids are amino acids that can be synthesized in the body; hence, they do not necessarily need to be part of the diet, but they are required for protein synthesis in the body. Examples include alanine, aspartic acid, and tyrosine [2]. Twenty amino acids are fundamental building blocks of proteins, out of which nine are essential (must be acquired from the diet). In comparison, 11 amino acids are non-essential (the human body can synthesize them).

In contrast, plants can produce all twenty amino acids. The nutritional quality of most conventional maize varieties is poor due to a deficiency in essential nutritional parameters [3]. The intake of these conventional varieties has led to undernutrition, especially in developing countries where maize serves as a staple food crop for most of the population [4]. The protein content of conventional maize is poor in quality because it contains small amounts of essential amino acids such as lysine (Lys) and threonine (Thr) [5]; this is also a significant setback in the nutritional quality of cereals [6]. The amino acids available in maize must be improved to militate against poor protein consumption in developing countries [5].

Biofortification has been used in recent times not just to improve the micronutrients level in crops but to enhance the overall nutritional benefits of crops. QPM (a biofortified maize variety) improves the nutritional status of the population that depends on maize as a staple crop [4], because diets with imbalanced amino acid levels contribute to the malnutrition conditions of Kwashiorkor in humans, while tryptophan’s deficiency can produce eye cataracts [2]. Quality protein maize (QPM), a biofortified *opaque-2* mutant maize variety, has special characteristic features such as low and uniform ear placement, resistance to ear rot, and root lodging. It also contains a notable amount of tryptophan, lysine, and protein, which are significantly different from the tryptophan, lysine and protein content of normal maize varieties. As a result, QPM significantly contributes to the nutritional diets of people of relatively low economic strength who depend on maize for their energy and protein intake. Its consumption can alleviate, to a very reasonable extent, the level of malnutrition in developing countries [1,3,7,8,9]. QPM varieties have been improved over the years to resist some disease attacks. Maize grain is classified as QPM if the quality index, the tryptophan-to-protein ratio in each sample, is higher than or equal to 0.008 [1]. The effectiveness of QPM in the growth of children in Ethiopia has been reported [4]. A significant 15% growth difference was observed between children fed with QPM and children fed with common maize for 13 months. In a similar study, children and infants fed with QPM had a 12% increase in the growth rate in weight and a 9% increase in the growth rate in height [10]. It was established that about 100 g of QPM is required for children to have enough lysine (the most limiting amino acid) in their diet [11].

Different methods have been used to quantify amino acids in crops for decades. A spectroscopic method for the rapid determination of amino acids has been developed [12]; High-Performance Liquid Chromatography (HPLC) methods have also been employed for amino acid quantification [4,13,14,15,16,17]; a Gas Chromatography-Mass Spectrometry (GC-MS) system has also been used to determine amino acids in corn seed. These methods have been sensitive and adequate for amino acid determination. However, they are all characterized by major setbacks, which include using expensive chemicals, tedious sample preparation to make the samples suitable for the instrumental analysis and being time consuming [18,19].

These setbacks disqualify these methods from being relatively economical and environmentally safe (due to the use of chemicals, which may be toxic) and from being high-throughput methods. Crop breeding using advanced genetic technologies has resulted in generating hundreds of clones that need to be screened for their nutritional traits in the shortest possible time.

NIRS is a rapid, low-cost, and environmentally friendly alternative to laborious laboratory analysis, which takes longer to complete and involves generating chemical waste. NIRS is a relatively inexpensive, rapid, non-destructive method that requires no or simple sample preparation to analyze target parameters in the samples of interest [20]. Previous studies have reported the application of NIRS to predict the nutritional composition of maize. NIRS calibrations were developed using more than 1100 samples collected over five years to predict methionine, cystine, lysine, threonine and tryptophan in cereals and middlings used for animal feed production [18]. In addition, the total contents of essential amino acids, protein, and moisture in protein-rich feed ingredients have been evaluated using NIRS models [19,21]. Furthermore, calibration models were developed for predicting tryptophan and lysine in whole maize grain, and the models showed good potential in segregating individual seeds [22]. However, lysine, methionine and other quality parameters in oats, barley, triticale and wheat across the Western Cape region of South Africa have been characterized using NIRS models [23], while in Mexico, calibration models were also developed for analyzing the amino acid content of pigmented maize samples planted across four locations [24].

However, only a few articles were published on NIRS to predict QPM’s amino acid (tryptophan and lysine) composition [20,25]. Therefore, this current study aims to develop prediction models for the characterization of amino acids in quality protein maize grown in Nigeria using NIRS.

## 2. Materials and Methods

### 2.1. Genetic Materials

A total of 83 quality protein maize samples provided by the Maize Improvement Program of IITA were used to develop calibration models for 17 amino acids, notably methionine, lysine, cysteine, and other essential amino and non-essential amino acids. The samples were split into 63 samples for calibration and 20 samples for validation sets using the WinISI Foss software.

### 2.2. Sampling for Laboratory Analysis

Dried maize grains were received in the laboratory, and a representative portion was selected from a bulk, sorted, and milled using a laboratory mill to fine particle size (<0.8 mm) and homogenized. A portion of the dried maize flour was transferred to a paper bag at room temperature before NIRS analysis. Another portion was collected and transferred immediately for HPLC analysis for reference data.

### 2.3. Spectra Data Collections and Pretreatments

Spectra information of each maize sample set was collected on the NIRS using a stationary ring samples cell. The samples were scanned in duplicate within the Vis-NIR wavelength range of 400–2498 nm, although only the NIR wavelength range (800–2400 nm) was used for multivariate analysis in this study, and the spectra data were reported as absorbance values log (1/R) at 0.5 nm intervals. Figure 1 shows a typical spectrum for the samples. The spectra were subjected to several pretreatments before calibration model development. Standard Normal Variate and De-trending (SNVD) and Multiple Partial Least Square (MPLS) were tested on two derivatives and three smoothing options. The treatment was represented by D, G, S1, and S2, where D indicates the derivative order number (0 indicates no derivation, 1 means the first derivative, and so on), G indicates the gap (the number of data points over which derivation is computed), S1 indicates the number of data points in the first smoothing (1 means no smoothing), and S2 indicates the number of data points in the second smoothing, where 1 means no smoothing [26,27]. The three pretreatment methods (SNVD+1,5,5,1, SNVD+ 2,5,5,1 and SNVD+2,10,10,1) were compared to identify the best treatment for better prediction models. Three cycles of outlier elimination were used. Samples with an H value (Mahalanobis distance) greater than 4 (spectral outliers) and a T value greater than 2.5 (samples that are unfit for calibration model) were eliminated in three cycles. The validation set was used to compare the predicted and the reference values using the WinISI Foss software.

### 2.4. Laboratory Analysis

The pre-column derivatization with 6-aminoquinolyl-N-hydroxy-succinimidyl carbamate (AQC) and High-Performance Liquid Chromatography (HPLC) and a fluorescent detection procedure were used in the analysis of the sample’s amino acid [28,29].

#### 2.4.1. Chemicals Used

Acetonitrile (HPLC super gradient grade) and methanol (HPLC super gradient grade) were purchased from Lab-Scan (Dublin, Ireland). Hydrochloric acid p.a. (36.5%) was a product of Ultrapure water produced by a Milli-Q Plus system (Millipore Corporation, Burlington, MA, USA). The AccQ•TagReagent Kit was purchased from Waters (Milford, MA, USA). The reagent kit consists of Waters AccQ•Fluor Borate Buffer, Waters AccQ•Fluor Reagent Powder (6-aminoquinolyl-N-hydroxy-succinimidyl carbamate—AQC), Waters AccQ•Fluor Reagent Diluent, Waters AccQ•Tag Amino Acid Analyzing Column (Nova-Pak C18, 4 µL, 150 × 3.9 mm), and Waters Amino Acid Hydrolysate Standard (each ampoule contains a 2.5 mM mixture of the 17 hydrolysate amino acids except for cystine—1.25 mM), i.e., aspartic acid (Asp), serine (Ser), glutamic acid (Glu), glycine (Gly), histidine (His), arginine (Arg), threonine (Thr), alanine (Ala), proline (Pro), cysteine (Cys), tyrosine (Tyr), valine (Val), methionine (Met), lysine (Lys), isoleucine (Ile), leucine (Leu), and phenylalanine (Phe)).

#### 2.4.2. Derivatization of the Hydrolysate

*Reconstituting AccQ•Fluor Reagent.* First, 1.0 mL of AccQ•Fluor Reagent Diluent was transferred into a vial containing Waters AccQ•Fluor Reagent Powder. This closed vial was mixed with Vortex (IKA^®^ Werke GmbH & Co. KG. Janke & Kunkel-Str. 10. D-79219 Staufen Germany/Deutschland) for 10 s and heated on a heating block (55 °C) until dissolving but for not longer than 10 min.

#### 2.4.3. Preparing a Calibration Standard

Internal standard method: The calibration standard solution was combined with an internal standard (6.45 mg α-aminobutyric acid to 25 mL 0.1 M HCl): 40 µL Amino Acid Hydrolysate + 40 µL internal standard stock solution and 920 µL Milli-Q water was transferred in the sample tube as a stock solution. Calibration curves were prepared using a serial dilution from the stock standard solution.

#### 2.4.4. Derivatizing the Calibration Standard

First, 10 µL of the standard calibration solution was transferred into the 6 × 10 mm sample tube, and 70 µL AccQ•Fluor Borate Buffer was added and vortexed for about 30–40 s using a Vortex Mixer (VELP Scientifica TX4 Digital IR Vortex mixer, Italy). This is followed by adding 20 µL, reconstituting AccQ•Fluor Reagent, and mixing immediately for several seconds. The content was then transferred to the bottom of low-volume insert vials and placed on a preheated heating block at 55 °C for 10 min. It was allowed to stand for a few minutes, and 5 µL of the derivatized standard was injected into the chromatographic system.

#### 2.4.5. Preparing Samples for AccQ•tag Method

Five micrograms of the pulverized sample were hydrolyzed with 5 mL of Constant Boiling 6 MHCL in a 10 mL hydrolysis tube using a CEM Microwave Discover Workstation. Each hydrolysate was centrifuged at 3000 rpm for 10 min to obtain a clear solution. Then, 10 µL of the diluted hydrolysate was pipetted into a 6 × 10 mm sample tube, and an equal volume of 0.1 M NaOH was added to neutralize the excess acid before derivatization. From this solution, 10 µL was taken for the derivatization procedure.

#### 2.4.6. Derivatization of Samples

First, 10 µL of the diluted hydrolysate + 70 µL AccQ•Fluor Borate Buffer were mixed in a sample tube and briefly homogenized with a Vortex mixer; then, 20 µL reconstituted AccQ•Fluor Reagent was added, and the mixture was mixed immediately for several seconds with vortex. The content was then transferred to an auto-sampler vial, and the vial was heated at 55 °C for 10 min.

#### 2.4.7. HPLC Analysis

A Waters Alliance 2695 HPLC system with a 2475 Multi λ Fluorescence detector (Waters, Milford, MA, USA) was used for the HPLC analysis (excitation at 250 nm and emission at 395 nm). An AccQ•Tag amino acid column Nova-Pak C 18, 4 µm (150 × 3.9 mm) from Waters was used. The column was thermostated at 37 °C, and 10 μL was the injection volume (concentration of amino acids 2.5–250 pmol). The mobile phase consisted of Eluent A (prepared from Waters AccQ•Tag Eluent A concentrate by adding 200 mL of concentrate to 2 L of Milli-Q water and mixing), Eluent B (acetonitrile, HPLC grade), and Eluent C (Milli-Q water). The gradient system used for the chromatography was as follows: The gradient separation program was as follows: Solvent A-100%, B-0%, C-0% runs from 0 to 0.5 min, A-99%, B-1%, C-0% flows from 0.5 min to 18 min of the runs, followed by A-95%, B-5%, C-0% for 1 min. Then, A-91%: B-9%, C-0% runs until 29.5 min of the analysis time and then A-83%, B-17%, C-0% runs to 33 min. A-0%, B-60%, C-40% runs for an additional 6 min and then completed at 36 min (A-100%, B-0%, C-0%). System equilibration was continued for another 10 min with 100% solvent A.

### 2.5. Calibration Models

The samples were split into 63 calibration sets and 20 validation sets using the WinISI Foss software. Prediction models for 17 amino acids were developed using three mathematical pretreatments of the spectra data. Multiple partial least squares (MPLS) regression and cross-validation techniques were used to calculate the correlation between spectral data and laboratory reference values for each spectrum. SNVD+ 1,5,5,1; SNVD+ 2,5,5,1; and SNVD+ 2,10,10,1 were the mathematical treatments used on the spectra data, and models from each treatment were compared to select the best prediction model. WinISI 4 project Manager software was employed to develop the calibration models. The selected 20 independent samples were used as the validation set to test the performance of the developed models.

## 3. Results and Discussion

Results of amino acids analysis (Table 1) showed that the following essential amino acids had a mean ± SD of 1.46 ± 1.87 for histidine (HIS), 0.85 ± 0.60 for threonine (THR), 2.01 ± 1.93 for valine (VAL), 0.31 ± 0.53 for methionine (MET), 0.91 ± 0.73 for lysine (LYS), 3.81 ± 1.76 for isoleucine (ILE), 6.10 ± 6.22 for leucine (LEU), and 3.20 ± 2.28 (g/100 g) for phenylalanine (PHE), respectively. Mean values of 0.18, 0.13, 0.25, and 0.54% for MET, LYS, THR, and LEU, respectively, have been reported [19]. These values are somewhat lower than those reported in this study, which reveals the amino acid quality of the maize samples analyzed. However, mean values of 2.95, 3.81, 3.68, and 2.20% for essential amino acids ILE, PHE, VAL, and HIS, respectively [30], agree with the values reported in this study. The recommended amounts of essential amino acids for infants are 47 mg/g, 29 mg/g, 52 mg/g, 35 mg/g, 80 mg/g, and 63 mg/g for VAL, MET, LYS, ILE, LEU, and PHE, respectively [31]. These limits are met by some of the QPM genotypes analyzed in this study, as the maximum values for these essential amino acids are 59.1 mg/g, 39.4 mg/g, 56.7 mg/g, 73.3 mg/g, 198.2 mg/g, and 80.1 mg/g, respectively, for VAL, MET, LYS, ILE, LEU, and PHE. (Table 1). Hence, these varieties can be used in formulating infants’ cereal meals.

The coefficient of determination in calibration (R^2^_cal_) for the developed model ranged from 0.07 to 0.93, with alanine and proline having the highest R^2^_cal_ of 0.93 and histidine having the lowest R^2^_cal_ (Table 2). The coefficient of determination in calibration in this study was extremely low for some amino acids, including the indispensable ones such as histidine, lysine, and methionine. This might be because the training data set does not represent the variance in the sample population for these amino acids [32]. However, some essential amino acids had a reasonably good R^2^_cal_ and low standard error of calibration (SEC), such as leucine (0.91), phenylalanine (0.75), and valines (0.79) with SEC values of 1.75, 1.12 and 0.87, respectively. Noel et al., 2021 [33] reported in their study on the prediction of protein and amino acids in cereals using NIRS coefficients of determination of 0.87, 0.93, and 0.96 for valines, phenylalanine, and leucine, respectively, with prediction performances of 0.87 for phenylalanine, 0.73 for valine, and 0.69 for leucine with SEP values of 0.40, 0.49 and 0.93. This current study reported R^2^_cal_ of 0.79, 0.75, and 0.91 and prediction performances (R^2^_pred_) of 0.82, 0.88, and 0.90 for valines, phenylalanine, and leucine with SEP values of 0.78, 0.77 and 2.20, respectively. The coefficient of determination in prediction and SEP in this study is slightly higher than reported by (33) for valines, phenylalanine and leucine, respectively. In addition, in the current study, the range of coefficient of determination in calibration (R^2^_cal_) of 0.51 to 0.93 accounts for about 70% of the amino acids, where aspartic acid had an R^2^_cal_ of 0.86, serine had an R^2^_cal_ of 0.71, glutamic acid had an R^2^_cal_ of 0.91, glycine had an R^2^_cal_ of 0.81, arginine had an R^2^_cal_ of 0.78, alanine and proline had an R^2^_cal_ of 0.93, valine had an R^2^_cal_ of 0.70, leucine had an R^2^_cal_ of 0.91, phenylalanine had an R^2^_cal_ of 0.75, threonine had an R^2^_cal_ of 0.51, and tyrosine had an R^2^_cal_ of 0.68. Their low SEC, which ranged from 0.22 to 1.75, shows the potential for improvement by carefully including wide variations in the training data set. An R^2^_cal_ of 0.96 was earlier reported for leucine [18], which is similar to the R^2^_cal_ reported in this study, but R^2^_cal_ values of 0.95 for valine, 0.90 for arginine, and 0.96 for threonine were reported [18], which are slightly higher than what was reported in our work. This could be due to the sample size (258 samples) used in the study [18], which may give better variability to the models developed against the 63 samples used in this report.

In addition, the R^2^_cal_ values for aspartic acid (0.52), glycine (0.59), glutamic acid (0.51), threonine (0.47), and valine (0.45) using the PLS regression method were published [24], which are all lower than the values reported in this study. The authors use whole grains instead of the ground grain used in the current study, which might be responsible for the low values. Many factors could be responsible for the very poor coefficient of determination in some of the essential amino acids in the current study, such as histidine (0.07), cysteine (0.13), methionine (0.09), lysine (0.20), and isoleucine (0.09), respectively. In addition to the lack of genetic variability of traits and poor representativeness of the samples within the calibration data sets, the accuracy of the reference method also affects the prediction performances of NIRS, which is the most critical factor that affects the reliability of NIRS calibration and prediction [34]. The reference method used in this study has a high sensitivity for both primary and secondary amino acids with high accuracy up to 50 ng [35].

The coefficients of determination in prediction for aspartic acid, arginine, alanine, and leucine were 0.90, 0.94, 0.90, and 0.89, respectively, which could be considered good and with a standard error of prediction (SEP) of 0.24, 0.24,0.93 and 2.20, respectively. In addition, all other amino acids have R^2^_pred_ above 0.60 except histidine (0.12), threonine (0.35), cysteine (0.18), methionine (0.08), lysine (0.20), and iso-leucine (0.13), respectively. Generally, for the NIR prediction to be suitable for rapid screening, the coefficient of determination in prediction should be in the range of 0.66 to 0.81. For quality control and accurate determination, it should have a range from 0.83 to 0.90 [32]. The model developed in this current study is sufficient for predicting some of the amino acids of QPM such as aspartic acid, serine, glutamic acid, arginine, alanine, proline, tyrosine, valines, leucine, and phenylalanine for rapid screening purposes. The author wishes to improve the prediction models using new genetic materials with wide genetic diversity in the training populations, especially for the other essential amino acids with poor prediction performances, such as lysine and methionine.

## 4. Conclusions

Quality protein maize (QPM) is preferred over traditional maize varieties due to its improved nutritional content, most notably the essential amino acid composition, which is an essential human growth factor. Applying near-infrared spectroscopy to characterize the amino acids in maize has helped to improve research efforts on breeding quality protein maize. Results from the current study have demonstrated the potential of NIRS prediction models to screen QPM for specific amino acids, including the essential ones such as arginine, leucine, phenylalanine, and valine. However, the prediction performances for some amino acids need to be improved by including QPM populations that have a wide variability for the amino acids in the training data set in our subsequent study. These models would serve as tools for maize breeding programs to rapidly screen their QPM germplasm for amino acids using the near-infrared spectrometer.

## Figures and Tables

**Figure 1 foods-11-02779-f001:**
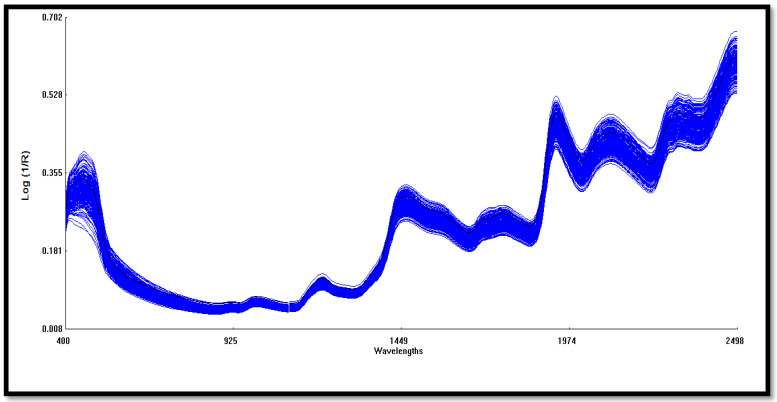
A typical average visible to the near-infrared (400–2500 nm) spectrum illustrates peaks of the calibration data set. The near infrared range (800–2400 nm) was used for the multivariate analysis.

**Table 1 foods-11-02779-t001:** Descriptive statistics summary of amino acids in QPM (N = 63).

Constituents	Minimum	Maximum	Mean	Standard Deviation
ASP	0.04	2.39	0.70	0.65
SER	0.13	2.85	0.98	0.72
GLU	0.06	7.12	2.28	2.12
GLY	0.02	2.68	0.80	0.76
HIS	0.04	10.38	1.46	1.87
ARG	0.01	6.50	0.88	1.05
THR	0.02	2.33	0.85	0.60
ALA	0.10	8.37	2.68	2.66
PRO	0.02	3.33	1.14	0.98
CYS	0.01	1.35	0.25	0.24
TYR	0.02	2.68	0.86	0.72
VAL	0.06	5.91	2.01	1.93
MET	0.02	3.94	0.31	0.53
LYS	0.07	5.67	0.91	0.73
ILE	0.23	7.33	3.81	1.76
LUE	0.51	19.82	6.10	6.22
PHE	0.51	8.01	3.20	2.28

N = number of samples.

**Table 2 foods-11-02779-t002:** Calibration and validation statistics of prediction models developed for amino acids. Pre-processing and mathematical treatments (SNVD, + 2,5,5,1).

	Calibration						Validation			
	N = 63					(N = 20; Outliers = 9)				
Constituent	SEC	R^2^_Cal_	SECV	Outliers	Pred	Lab	SEP	Bias	Slope	R^2^_pred_
**ASP**	0.22	0.86	0.37	4	0.65	0.52	0.24	0.13	0.24	0.90
**SER**	0.35	0.71	0.55	6	1.08	0.76	0.49	0.32	1.01	0.61
**GLU**	0.62	0.91	1.17	4	0.76	1.96	1.16	−0.32	1.24	0.70
**GLY**	0.30	0.81	0.45	4	0.75	0.65	0.39	0.17	0.39	0.80
**HIS**	1.00	0.07	1.87	4	2.01	1.12	2.22	0.88	3.12	0.12
**ARG**	0.32	0.78	0.95	5	0.70	0.50	0.24	0.14	1.18	0.94
**THR**	0.35	0.51	0.53	6	0.95	0.65	0.52	0.29	1.04	0.35
**ALA**	0.71	0.93	1.27	3	2.17	2.46	0.93	−0.28	0.99	0.90
**PRO**	0.26	0.93	0.48	3	1.07	1.09	0.47	−0.02	0.79	0.80
**CYS**	0.09	0.13	0.23	5	0.22	0.22	0.08	0.03	0.18	0.18
**TYR**	0.37	0.68	0.55	5	0.76	0.69	0.34	0.13	1.24	0.83
**VAL**	0.87	0.79	1.31	3	2.05	1.57	0.78	0.48	1.03	0.82
**MET**	0.14	0.09	0.54	5	0.10	0.20	0.14	−0.02	−0.58	0.08
**LYS**	0.36	0.20	0.72	1	1.02	0.88	0.40	0.13	0.73	0.20
**ILE**	1.68	0.09	1.78	0	4.43	3.74	0.84	0.68	0.84	0.13
**LUE**	1.75	0.91	2.88	4	5.19	4.34	2.20	0.84	2.20	0.90
**PHE**	1.12	0.75	1.45	2	2.70	2.66	0.77	0.10	1.16	0.88

R^2^_cal_ = coefficient of determination in calibration, SEC = standard error of calibration, SECV = standard error of cross-validation; SEP = standard error of prediction, R^2^_pred_ = coefficient of determination in validation, Np = number of samples used for prediction, Nc = number of samples used for the calibration; Pred = predicted values, Lab = wet analysis values; Outliers: Samples eliminated with H value (Mahalanobis distance) greater than 4.

## Data Availability

The data presented in this study are available on request from the corresponding author.

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
