# Peer review of "Near-Infrared Reflectance Spectrophotometry (NIRS) Application in the Amino Acid Profiling of Quality Protein Maize (QPM)"

_foods, 2022, doi:10.3390/foods11182779_

Round 1
Reviewer 1 Report
This work took a challenging aim, as it is known that quantitative analysis of aminoacids by spectroscopy is often difficult. However, the results presented in this work indicate that no reliable and accurate analysis of aminoacid contents in maize was successful.
This, unfortunately, makes it particularly important to carefully design the experiment. As Table 2 clearly presents, the models trained for the dataset might be overfitting - the calibration set was too scarce for fully reliable calibration. The large differences between SECV and SEC values, and in most cases SEP values being lower than SECV values, indicate that the calibration set was probably not representative well-enough for the variance in the population. This might be not that critical for drawing rough conclusions, but should be addressed in the text. Moreover, CYS model is clearly of very poor reliability (SECV – 0.23; SEP – 0.06; this suggests that the modelling was sensitive to which samples specifically were assigned to training and validation sets) and authors could take a closer look into their data during revision.
Another problem is, the Authors sometimes make attempt to present these results in a much better light than they ought to be. This is potentially deceptive towards the readership perhaps less familiar with chemometrics and analytical spectroscopy. For example, the sentence:
“The models for GLU, GLY, ALA, PRO, TYR, VAL, and LEU have good predictive performance with R2 of 0.54, 0.51, 0.58, 0.55, 0.54, 0.56, and 0.68, respectively.”
The R2 values at the level of 0.5-0.6 can, by no means, be described as indicative to a good predictive performance. In fact, this is a very poor model fit, well-below the acceptable standards. Typically, even in challenging analyses such as natural products analysis, agricultural items, etc. the R2 of at least 0.8 is considered sufficient for “rough screening” (a vast amount of literature and practical knowledge was gathered in these areas, including grain analysis, and therefore, very-well established thresholds for reliability of models are commonly accepted in the community).
My suggestion is that the authors improve the discussion also including a fair assessment of the quality of the obtained results with literature, to specifically avoid the potential misleading. Some recent literature published in Foods journal that provides good background and overview of the performance metrics in NIR spectroscopy and modelling quantitative properties of agri-food items would be helpful for the readership (e.g. DOI: 10.3390/foods11101465).
Author Response
Thank you for your in-depth review of the manuscripts, technical observations, and suggestions. We have rewritten the discussion of the manuscripts with references to other published literature to improve the discussion in line with your observations and remarks. Kindly see all the additions in track changes in the revised manuscript.

Reviewer 2 Report
The theme of the manuscript is very interesting. However, the manuscript is not written with care! The evaluation of the spectra is not described in a proper way (For example: How many principal components are used?). Furthermore, it is not clear how the samples are gathered.
Please write Cysteine, Methionine, Lysins, and Isoleucine not with capital letters?
Why the tryptophan-to-protein ratio is given as a percentage value such as 0.8 %, better is 0.008.
The preparation of the samples is unclear. A bulk was available and randomly samples were selected. So two or more samples could be the same?
NIR is not in the range of 400 – 2498 nm! If the full range is used, then change the title.
“1 means no smoothing” is presented twice. If there is a difference, please explain.
What is meant by “+1551” and “+2552” and “+210101”?
Figure 1: Units are missing.
Is cysteine or cystine determined?
3. Result and Discussion: Please do not repeat the same results, which are presented in Table 1, without any comment.
What are the units in Table 1?
Table 1: N is always the same, just mention it in the table caption.
Values always must have units!
Table 2: What is N if just 20 samples are in the prediction set. What are the units? What is meant by Math? If Scatter, Math and Regression method is always the same, it should be just mentioned in the table caption.
To judge the results better, a percentage error (=standard error divided by range) should be calculated and this value as well as the SEP should be presented.
How good can the amino acid be determined using the reference methods? This must be mentioned.
Author Response
Reviewer 2
Comments and Suggestions for Authors
The theme of the manuscript is very interesting. However, the manuscript is not written with care! The evaluation of the spectra is not described in a proper way (For example: How many principal components are used?). Furthermore, it is not clear how the samples are gathered.
Please write Cysteine, Methionine, Lysins, and Isoleucine not with capital letters?
Response: Cysteine, Methionine, Lysine, and Isoleucine have been written lower case in the text as advised.
Why the tryptophan-to-protein ratio is given as a percentage value such as 0.8 %, better is 0.008.
Response: This has been rewritten in the text
The preparation of the samples is unclear. A bulk was available and randomly samples were selected. So two or more samples could be the same?
Response: Clear information on the sampling and sample preparation was pointed out under the materials and methods section.
NIR is not in the range of 400 – 2498 nm! If the full range is used, then change the title.
Response: The range used was 400-2498nm (the range of the FOSS equipment used to collect the spectra data).
“1 means no smoothing” is presented twice. If there is a difference, please explain.
Response: Repeated sentences have been removed from the text
What is meant by “+1551” and “+2552” and “+210101”?
Response: Repeated sentences have been removed from the text
Figure 1: Units are missing.
Is cysteine or cystine determined?
Response: Units have been included in Figure 1. Cysteine was determined
- Result and Discussion: Please do not repeat the same results, which are presented in Table 1, without any comment.
Response: This was corrected in the presentation of the results as advised
What are the units in Table 1?
Table 1: N is always the same, just mention it in the table caption.
Values always must have units!
Response: The table has been rewritten N is removed, and the unit of quantification was added
Table 2: What is N if just 20 samples are in the prediction set? What are the units? What is meant by Math? If Scatter, Math and Regression method is always the same, it should be just mentioned in the table caption.
Response: This has been corrected accordingly in Table 2
To judge the results better, a percentage error (=standard error divided by range) should be calculated and this value as well as the SEP should be presented.
Response: Standard error of prediction was provided in the statistics in Table 2
How good can the amino acid be determined using the reference methods? This must be mentioned.
Response: The HPLC Fluorescence detector coupled with pre-column derivatization using 6-amino quinoline succinimidyl carbamate is reliable for amino acids and has high sensitivity up to 50 ng. This point has been mentioned in the discussion sections of the text.
We sincerely appreciated your comments

Reviewer 3 Report
“Near-Infrared Reflectance Spectrophotometry (NIRS) application in the amino acid profiling of Quality Protein Maize (QPM)” by E. O. Alamu et. al. describes the use of NIR to quantitate the amino acid content in maize as an alternative to more cumbersome techniques such as HPLC. To accomplish this, the authors built models using 63 different maize genotypes for the calibration set and 20 for the validation set to correlate with HPLC results for the same samples.
In the abstract the authors only mention the R2 values for 16 of the 17 amino acids that were assessed in the main body of the text, Threonine is missing.
In the Results and Discussion section the authors go back and forth comparing obtained values with literature values for various amino acid concentrations without being clear when they are referring to data they obtained vs literature data. This can be greatly cleaned up for readability.
The authors state that “However, R2 for aspartic acid (0.52), glycine (0.59), glutamic acid (0.51), threonine (0.47), and valine (0.45) using PLS regression method were published [24], which are all lower than the values reported in this study. This indicates that the regression method (MPLS) used in this study gave better calibration models than PLS even though 143 samples were used [24].” However in reviewing reference 24 the authors fail to note that the authors in reference 24 using whole grains (vs ground used in this submitted publication) which will have a determinantal affect on data quality at the expense of being a much faster approach. Also if the authors want to compare model types they should simply run PLS regression models on their own data to see which gives the better results, the conclusion can’t be drawn from comparing to a different data set. Additionally, the authors would benefit from generating visuals such as the correlation heat maps and network maps to better illustrate and explain trends observed in their data.
Was any evaluation of model robustness performed? i.e. different analysts performing measurements, performed on different days from the generation of the calibration set. This is important in having a useful model since small variations such as daily temperature, humidity, analyst technique, etc. can have an affect on model performance and would be present if this approach was actually being used as a screening tool.
The authors note that Lysine is very important since it is the most limiting amino acid in childrens diets and also highlight the importance of threonine in the introduction. However the predictive performance for these two ranges from very poor (for lysine) to poor (for threonine). Would HPLC or another type of test still have to be performed to screen for these amino acids in this situation?
Optimization of the pre-processing and model parameters is an important step, but we don’t need tables in the body of the text showing the comparison. This can be moved to the SI or removed completely.
Overall, this manuscript as written doesn’t show anything new or noteworthy in for amino acid profiling of QPM. Additional model development should be addressed including model robustness and interpreting model loadings. The results and discussion should be rewritten to highlight the data obtained by the authors and not confusing comparisons between values from other manuscripts.
Author Response
Reviewer 3
Comments and Suggestions for Authors
“Near-Infrared Reflectance Spectrophotometry (NIRS) application in the amino acid profiling of Quality Protein Maize (QPM)” by E. O. Alamu et. al. describes the use of NIR to quantitate the amino acid content in maize as an alternative to more cumbersome techniques such as HPLC. To accomplish this, the authors-built models using 63 different maize genotypes for the calibration set and 20 for the validation set to correlate with HPLC results for the same samples.
In the abstract the authors only mention the R2 values for 16 of the 17 amino acids that were assessed in the main body of the text, threonine is missing.
Response: Thank you for your observations. The results for threonine have been included in the main text
In the Results and Discussion section, the authors go back and forth comparing obtained values with literature values for various amino acid concentrations without being clear when they are referring to data they obtained vs literature data. This can be greatly cleaned up for readability.
Response: The discussion part of the manuscript was rewritten to improve the readability
The authors state that “However, R2 for aspartic acid (0.52), glycine (0.59), glutamic acid (0.51), threonine (0.47), and valine (0.45) using PLS regression method were published [24], which are all lower than the values reported in this study. This indicates that the regression method (MPLS) used in this study gave better calibration models than PLS even though 143 samples were used [24].” However in reviewing reference 24 the authors fail to note that the authors in reference 24 using whole grains (vs ground used in this submitted publication) which will have a determinantal affect on data quality at the expense of being a much faster approach. Also if the authors want to compare model types they should simply run PLS regression models on their own data to see which gives the better results, the conclusion can’t be drawn from comparing to a different data set. Additionally, the authors would benefit from generating visuals such as the correlation heat maps and network maps to better illustrate and explain trends observed in their data.
Response: Thank you for this important observation in the manuscripts. We have also realized this and corrected it accordingly in the manuscript. We hope that the revised copy of the manuscript has improved.
Was any evaluation of model robustness performed? i.e. different analysts performing measurements, performed on different days from the generation of the calibration set. This is important in having a useful model since small variations such as daily temperature, humidity, analyst technique, etc. can have an affect on model performance and would be present if this approach was actually being used as a screening tool.
Response: We did not conduct separate tests for model robustness, but an independent set of samples was only used to test the performance of the calibration model, and the prediction regression was reported. However, we are using the models for the current season harvest to establish the robustness and possibly improve the models using new samples.
The authors note that lysine is very important since it is the most limiting amino acid in childrens diets and also highlight the importance of threonine in the introduction. However the predictive performance for these two ranges from very poor (for lysine) to poor (for threonine). Would HPLC or another type of test still have to be performed to screen for these amino acids in this situation?
Response: A few publications have shown the potential of NIRS to provide good prediction performances for lysine. Though that was not the case in our study, we hope to improve further the selection of our training data set to include wide genetic diversity to increase the variability of the amino acids in the sample.
Optimization of the pre-processing and model parameters is an important step, but we don’t need tables in the body of the text showing the comparison. This can be moved to the SI or removed completely.
Response: The preprocessing and model parameters information has been removed from the main text as advised.
Overall, this manuscript as written doesn’t show anything new or noteworthy in for amino acid profiling of QPM. Additional model development should be addressed including model robustness and interpreting model loadings. The results and discussion should be rewritten to highlight the data obtained by the authors and not confusing comparisons between values from other manuscripts.
Response: Thank you once again for your comments. We have rewritten the discussion part of the manuscripts. We do not focus on identifying new amino acids in QPM. Still, we are looking at developing models for amino acid measurements using a high throughput technique that will be cost-effective and efficient. We believe we have succeeded in establishing relative good models for the majority of the amino acids. As mentioned earlier, we are currently using the models to profile this season’s harvest to establish robustness and improve the models.

Round 2
Reviewer 3 Report
While the authors have addressed many of the initial concerns regarding the manuscript they have also raised more. As the added text below states, only leucine would fall within the range of being suitable for NIR prediction for rapid screening.
"Prediction performances for all the amino acids were slightly above 50%, with leucine having the highest value of 0.68 and iso-leucine having the least (0.01) value. Generally, for the NIR prediction to be suitable for rapid screening, the coefficient of determination in prediction should be in the range of 0.66 to 0.81."
Additionally the RSQ for prediction using the selected prediction models falls drastically from the RSQ from calibration often times by half and in some cases even more drastically such as SER where it goes from a RSQ of 0.71 for calibration to 0.08 for RSQ of prediction. Since the validation set is just a randomly selected subset of the initial data this indicates that the model is very overfit and not useful for predictive performance.
While re-running the experiments may be cumbersome, it should be feasible for the authors to try different approaches to model building on the already obtained spectral data as well as different model types to see if the predictive performance of the models can be improved. Otherwise the drastic drop-off in RSQ from calibration to validation indicates these models wouldn't be useful for a rapid screening environment.
